# Azo-Dyes-Grafted Oligosaccharides—From Synthesis to Applications

**DOI:** 10.3390/molecules26113063

**Published:** 2021-05-21

**Authors:** Estelle Léonard, Antoine Fayeulle

**Affiliations:** Université de Technologie de Compiègne, ESCOM, TIMR (Integrated Transformations of Renewable Matter), Centre de Recherche Royallieu, CS 60 319, CEDEX, 60203 Compiègne, France; antoine.fayeulle@utc.fr

**Keywords:** azobenzene, oligosaccharides, cellulose, cyclodextrin, host/guest, grafted photochromes

## Abstract

Azobenzenes are photochromic molecules that possess a large range of applications. Their syntheses are usually simple and fast, and their purifications can be easy to perform. Oligosaccharide is also a wide family of biopolymer constituted of linear chain of saccharides. It can be extracted from biomass, as for cellulose, being the principal constituent of plant cell wall, or it can be enzymatically produced as for cyclodextrins, having properties not far from cellulose. Combining these two materials families can afford interesting applications such as controlled drug-release systems, photochromic liquid crystals, photoresponsive films or even fluorescent indicators. This review will compile the different syntheses of azo-dyes-grafted oligosaccharides, and will show their various applications.

## 1. Introduction

### 1.1. Azo Molecules

Photochromic family is composed of several switches, which are able to change their conformation upon light illumination. Some molecules can be cited as spiropyrans [1], chromenes [2,3] indogoid [4] and azobenzenes [5], also known as diazo molecules.

Diazo molecules are photochromes which are capable of an isomerization under a UV-light illumination, leading to their stable *trans*-form to their metastable *cis*-form (Figure 1) [6].

A simplified state model for azobenzene isomerization can be described, leading to the energy that the system needs or release during the transformations (Figure 1) [7]. And for *cis*- to *trans*-thermal release, the azobenzene substituents as well as the electronic distribution allow the thermodynamically stable *trans*-form of the azo-dye to be restored very quickly [8].

Two ways of isomerization can be discussed, leading from *trans*- to *cis*-isomer: N=N torsion (rotamers) vs. N-inversion (invertomers) [9,10,11,12,13,14,15]. However, according the review of Bandara and Burdette [16], if trans-azobenzene isomerization always occurs in the S1 state (Figure 2) by an inversion process, regardless of the initial excitement, some theoretical studies predict isomerization of trans-azobenzene to be multi-dimensional or dominated by rotation.

Amongst the possibilities offered for their syntheses, several pathways can be found in the literature (Figure 3) [17].

Diazo coupling which implies the formation of a diazonium salt attacked by an electron-enriched aromaric.Mills reaction or oxidative/reductive coupling for in situ the formation of the diazo bond between two anilines/nitrobenzenes.

For the diazo coupling, several phenolic compounds can provide good electron-enriched substrates. We can cite for example the nude phenol [18,19,20,21,22,23,24], but also natural compound such as salicylaldehyde [25,26,27], vanillin [28,29], or even resorcinol [30,31].

Oxidative/Reductive couplings lead to symmetrical azobenzenes. For symmetrical reductive couplings, from nitrobenzene derivatives, reductive metals can be used such as Zinc [32] or Magnesium [33]. Catalytic hydrogenation can also successfully be used [34,35], as well as complex hydrides [36]. For symmetrical oxidative couplings, KMnO_4_/CuSO_4_, MnO_2_ or *m*CPBA can be used [37,38,39,40]. However, the simplest way for the symmetrical oxidative coupling is to use the air oxygen [41,42,43,44]. Mills reaction allows also oxidative coupling from anilines, but leading to unsymmetrical azobenzenes [45,46,47,48,49,50].

Many applications can be found for molecules bearing an azo moiety. If dyes as application are the most popular [51,52,53,54], many examples can be found for antimicrobial effect, where *trans-cis* isomerization impact on bacteria were discussed [55,56,57], for retina recovery [58,59], or even in the electric domain, by combining azobenzenes with triphenylamine moieties, where ROMP (Ring-Opening Metathesis Polymerization) thus used led to electrical storage capability polymers [60] or when azobenzenes are co-sensitizers for solar cells [61]. In the domain of synthetic enhancement, azobenzene can also be used as chemoreactor for catalysis in water [62,63] or enhancement of cellulolytic enzyme for example [64].

Even if this molecule can be toxic depending on what group is present on the aromatic rings [65] the use of such a photochrome is of great interest.

### 1.2. Oligosaccharides

All saccharide molecules are characterized by different factors, such as their anomeric configuration (α, β), their series (d, l), their relative configuration (gluco-, galacto- etc.) and their cycle form (furanose/pyranose). When the number (n) of glycosidic moieties between monosaccharides varies between 2 and 10, by convention, the formed oligomer is called oligosaccharide. However, this osidic number is not always fixed, and bigger oligosaccharides can be found. Indeed, polysaccharides are molecules with a degree of polymerization higher than 20–25 but this definition is not so strict, as with the presence of 25 residues, they can still be called oligosaccharides [66]. That is why small cellulose can be classified as oligosaccharide.

When n = 2, the well-known saccharose or cellobiose are obtained for example by starch degradation [67,68] and when n = 3, we can find isomaltotriose, raffinose or melecitose (Figure 4), found in 1833 from the larch tree [69] and also present in honey for example. If n = 4, formula can lead for example to the stachyose oligosaccharide (Figure 5).

Oligosaccharides can also be linear (Stachyose) or cyclic (Cyclodextrins), where n = 7 for the *beta*-version (presented in Figure 5), and with a lampshade-shape ring [70].

Amongst linear oligosaccharides, the most known contain a sequence of monomeric sugars and their name are related to these sequences [71]. We can cite for example fructooligosaccharides (FOS), galactooligosaccharides (GOS), xylooligosaccharides (XOS) shown in Figure 6, but also arabinooligosaccharides (AOS), and algae derived marine oligosaccharides (ADMO) from which Chitosan is of growing interest [72].

Cyclodextrins are cyclic oligosaccharides composed of α-(1→4)-linked glucosyl units. They possess a central hydrophobic cavity, whereas the outside of the lampshade is hydrophilic [73]. There are three major versions of cyclodextrins: alpha (where n = 6 glucose units), beta (where n = 7) and gamma (where n = 8). For these three objects, the cyclodextrins possess cavities even as isolated molecules in the absence of guests. All cyclodextrins have a height of 7.9 Å and an outer diameter of 14.5–17.5 Å. The diameter of the hydrophobic cavity is around 4.9 Å for α-, 6.2 Å for β-, and 7.9 Å for γ-cyclodextrin, where some variations can apply (Figure 7) [74].

Cyclodextrins possess also two faces: the primary face (shortest) and the secondary face (largest).

With this special shape and physicochemical properties, cyclodextrins can host organic compounds and solubilize them in aqueous solutions. This can lead to various applications, such as odor control [75] separation and analysis [76] catalysis [77,78] or even drug delivery [79].

So the combination of azobenzene moieties and small or large oligosaccharides, linear or cyclic can lead to great applications and this is what will be seen in the next parts of this review.

## 2. Mono/Multivalent Sugar Photoswitch

Azobenzene can be attached to one or more oligosaccharides leading to a mono or multivalent sugar photoswitch mainly dedicated to lectin-based adhesion control.

Indeed, the inhibition of bacterial adhesion on surface if of great interest. Often bacterial adhesion depends on the interaction of adhesive organelles called fimbriae. For example, in the case of *Escherichia coli*, this bacterium uses fimbriae (long, hairlike organelles that project from the bacterium’s surface) to establish infection (Figure 8) [80]. They consist of interlinking subunits of a single protein called pilin that forms a rigid, coiled helix-shaped rod. Sticky proteins called adhesins cap the tip of the rod and bind to carbohydrate receptors on their host.

By designing antagonists of the respective carbohydrate-bacterial lectin, this can lead to an adhesion inhibition [81]. That is what was done by V. Chandrasekaran et al. in 2013 where the first azobenzene mannobioside as photoswitchable ligand for the bacterial lectin FimH was synthesized [82]. First the azobenzene mannoside was synthesized followed by a second glycosylation using a mannosyl donor (Figure 9). It is noticeable that both *trans*- and *cis*-conformations of the tested molecule had an 50% inhibitory of surface adhesion on mannan-coated surface similar to the power the best standard against *E. coli* (*p*-nitrophenyl α-d-mannoside).

Other azobenzene-conjugated carbohydrates were synthesized, and led to hydrogelator capacity with specific affinity for lectins. The derivatives were lactonolactone-glycine-azobenzene, maltonolactone-glycine-azobenzene or cellobionolactone-glycine-azobenzene, where lactonolactone-glycine-azobenzene was shown to provide a bioactive interface for cell attachment [83].

When azobenzenes containing multivalent sugar ligands were synthesized (Figure 10), these azobenzene-appended sugar derivatives showed high affinity binding with the relevant lectins, especially with a high increase in binding affinity related to the monomeric sugar ligand alone. More interestingly, isomerization of the *cis*-form led to a better binding with the lectin than the *trans*-form [84].

Early studies of tris-diazo-cellobiose called phlorocello have also already been done for immunochemical tests (Figure 11). This study made in 1965 [85] had already in mind that mono- and disaccharides are antigenic when conjugated to a protein carrier and that antibodies produced show a specificity for the introduced sugar.

## 3. Cellulose

### 3.1. Azo-Dye-Grafted or Coupled Cellulose

Azobenzene introduction onto cellulose molecular chains through etherification reaction display reversible *trans-cis-trans* photoisomerization under successive UV/vis illumination [86]. This interesting coupling involving a non-labile bond and synthesized easily from an epoxyazobenzene can lead to good solubility of the final polymer, as well as a good resistance against the phenomenon of fatigue.

Esterification can also lead to azobenzene-functionalized cellulose. The synthesis implies a conventional DCC/DMAP activation of the reactants (Figure 12) for the ester formation from hydroxypropylcellulose (HPC) [87].

Analyses made for the AZO-HPC characterization were conventional. Indeed, if the ester and amide groups were easily seen on infrared spectra (1731 and 1695 cm^−1^), the proton NMR led to the calculation of the degree of substitution DS(AZO) using the formula:DS(azo)=13∑A(HPC)÷(7+6×DS(HPC))∑A(AZO)
where ∑A(AZO) is the sum of the integrated areas of the peaks from the azobenzene moieties at 7.82, 7.68, 7.41 and 2.69 ppm;

∑A(HPC) is the sum of integrated areas of the peaks of HPC at 3–5.5, 1.20 and 1.07 ppm;

DS(HPC)=3.57 is the degree of substitution with the hydroxypropyl groups of HPC.

For smart fibrous materials, the functionalization of commercial microcrystalline cellulose with azobenzene molecules can be done, leading to photo- and thermal-responsive materials. Moreover, nanoporous and non-porous nano/micro fibrous materials can be made by electrospinning of azobenzene-cellulose solutions, leading to fibrous materials, with light-reversible functional groups [88].

### 3.2. For Drug Delivery and Health

Host-guest interactions play an important role in the cyclodextrin (CD) area. Indeed, CD dimers can interact with hosts [77,89] and this interaction can have an important physicochemical involvement. Kim et al. in 2020 synthesized azobenzene-grafted carboxymethyl cellulose (CMC) hydrogels for a controlled drug release system [90]. This system was photo-switchable, reduction-responsive and self-healing as it is seen in Figure 13. Moreover, as the system was found to be non-cytotoxic, a naproxen (non-steroidal anti-inflammatory drug) release of the hydrogels could be photo-controlled to be able to deliver up to 80% of drug within 3 h by UV light or reducing agent.

### 3.3. With Liquid or Nano Crystal Properties

Azobenzenes can also be covalently linked to hydroxypropyl cellulose (HPC) [91]. Then, AZO-EHPC [92] or AZO-HPHPC (Figure 14) can give access to a degree of azobenzene substitution in the range of 0.6–1.8 by adjusting the mole ratio of azobenzene to HPC. They also had the property of a very low Tg of approximately 20–40 °C being thermotropic liquid crystals and also fully reversible with a *trans-cis-trans* transition upon alternating irradiation of UV and visible light. However, the mixing of heat/cooling and UV illumination were not tested, maybe because of the thermal instability of the metastable *cis*-azobenzene.

Smart responsive nanomaterials have received much attention over the last decade for their functional properties in response to environmental variables and especially light [93,94]. In this context, exploring photochromic cellulose nanocrystals (CNCs) functionalized thanks to poly {6-[4-(4-methoxyphenylazo) phenoxy]hexyl methacrylate} (PMMAZO) can lead to a changing of color in UV irradiation or pH, and thus to a great probe related to these stimuli (Figure 15). It is also noticeable that for UV/Vis illumination, the process was claimed to be fully reversible without fatigue [95].

### 3.4. For UV Protection

Thanks to the ozone layer, from the broad irradiation emitted by the sun, only UVA and UVB reach the earth’s surface. By functionalizing cellulose fabrics, one can induce UV protection, which is becoming of great interest. Azobenzenic Schiff bases were synthesized and coupled on cellulose with two concentrations of 2 g/L (called FC-1) and 5 g/L (called FC-2). The results are of great interest, as the UV protective properties of fabrics can be evaluated as good when the ultraviolet transmittance T(UVA) or T(UVB) is less than 5% and ultraviolet protection factor (UPF) reaches 10 to 30 (Table 1) [96].

The UV-protective properties of the functional cellulose fabrics were mainly attributed to absorbing UV radiation of azobenzene Schiff base, which caused change of the molecular structure based on its *cis-trans* isomerization and intermolecular proton transfer.

## 4. Cyclodextrins

### 4.1. Inclusion/Exclusion of Azobenzenes

Cyclodextrins are well-known for their capability to include in their cavity hydrophobic compounds and especially azobenzenes [97,98,99,100,101]. But they can also be chemoreactors for diazo coupling. This is what M. Craig et al. did in 1999 where azo-rotaxane was synthesized as a single isomer (Figure 16) [102].

But more surprisingly, a *trans*-azobenzene can inhibit the catalytic hydrolysis of an ester group by a cyclodextrin [103]. Indeed, a *trans*-azobenzene, can be included in the cyclodextrin cavity. However, when illuminated at 320–390 nm, the isomerization to *cis*-azobenzene force it to go out of the cavity, letting place for the ester group to enter the cyclodextrin, and to de hydrolyzed (Figure 17).

### 4.2. Azo Functionalised Cyclodextrins

Azo-modified cyclodextrins on their primary face have been investigated by two approaches. The first one was a nucleophilic substitution with a mono-tosylated cyclodextrin by a hydroxyazobenzene. The second one was done by click chemistry (Figure 18) [104].

However, the face to be modified has a great importance. Indeed, Ma et al. in 2007 proved that [1] rotaxane functionalized in position 2 were found to be different from the corresponding isomers on the 6 position (Figure 19). More detailed, they have a better aqueous solubility, induced circular dichroism and different absorptions [105].

The ways of synthesis play also a great role as azo-modified cyclodextrins can self lock/unlock [106]. Indeed, hydrothermal coupling between azidocyclodextrin and propargylic azobenzene lead to self-locked molecule which can self-assemble in bimolecular capsule while plunged in water, reversibly by adding DMSO. However, Huisgen 1,3-dipolar cycloaddition leads to self-unlocked molecule, which can self-assemble in linear supramolecule (Figure 20).

Moreover, it was proven that an azobenzene bearing two cyclodextrins was still able to isomerize properly, as shown in 1998 by Aoyagi et al. (Figure 21) [107].

The value of pH can also strongly modify the absorption of azo-modified cyclodextrin [108]. That was demonstrated in 1997 by Aoyagi et al. [109].

Here can be seen (Figure 22) a strong shift of the maximal absorbance of the molecule. For example, at a pH of 2.75, this maximum (corresponding to the π→π* of the hole trans molecule) is around 370 nm, whereas at a pH of 11.05, this maximum had shift to 490 nm with a hyperchromic effect. The shift is nearly the same for the absorbance at 270 nm (corresponding to the π→π* of the aromatic rings) shifted to 300 nm.

In terms of applications, azo-modified cyclodextrins can act as sensor or binder for organic molecules [110,111,112]. For example, azo-modified β-Cyclodextrin polymer can be used as a sensor for chlorophenols in water. Indeed, chlorinated by-products (CBPs) are formed in water as a result of the reaction of chlorine and its derivatives, used in the disinfection of water, with natural organic matter (NOM). So the sensitive detection of such pollutants is of great interest, especially in drinking water [113]. The sensor was synthesized as seen on Figure 23, and the 2,4-dichlorophenol was used as pollutant model, showing a good sensitivity factor compared to chlorophenol, and phenol.

Another fluorescent probe for the detection of cholic acid and its derivatives (litho-, deoxy-, chenodeoxy-, ursodeoxy-, hyodeoxy-cholic acid) was obtained by linking bis dansyl-modified β-cyclodextrin dimer with azobenzene, and showing a selective molecular recognition for steroidal compounds. Indeed, in this study, the dansyl moiety moves out of the cyclodextrin cavity upon guest binding and play a role as a hydrophobic cap, showing a decrease of the fluorescence [114].

In the domain of amphiphiles, the *trans-cis* isomerization can lead to a great difference of autoassembling, thanks to a host-guest (de)complexing. That was demonstrated by Guo et al. in 2019, where a photoresponsive self-assembly system based on host-guest inclusion was synthesized and tested upon illumination [115]. An azobenzene group was linked to a β-cyclodextrin by an alkyl chain, generating a novel amphiphile which showed reversible aggregate behavior due to photo-isomerization of azobenzene. In the case of *trans*-form, intermolecular host-guest complexes between CD and azobenzene were formed, and further self-assembled into curved linear structures in water. Dissociation of *cis*-azobenzene group from CD cavity was achieved upon illuminated by UV light, leading to the aggregate transformation into vesicles (Figure 24).

For *trans*-molecule, intermolecular host-guest recognition occurs, and curved linear structures are observed, which are mainly driven by hydrogen bonding. Upon UV light, *cis*-isomer disfavors host-guest complex formation, and acts like a conventional amphiphile generating vesicles (Figure 25).

## 5. Conclusions

To conclude, the azo-functionalization of natural and unnatural oligosaccharides leads to a high diversity of properties of interest.

First, the azo moieties can be attached to linear, big, small or cyclic sugars. This versatility is of great interest from a synthetic point of view (Table 2). Indeed, the links between azo moieties and oligosaccharides are usually ethers or amides between an already formed azobenzene. However, diazotation can be made in situ on aminophenyl cellobioside for example.

Secondly, the applications are various (Table 3), such as for health (1965 to 2013), hydrogels and liquid crystals (2013 to 2020), self assemblies (1999-2019) or sensors (1979 to 2011).

Finally, azo molecules, depending on the starting oligosaccharide material, can act as a functional group, or a guest, leading to unprecedented physicochemical aspects even if sometimes, the UV impact on the structure could have been investigated deeply. So linking an azobenzene group to an oligosaccharide is simple to synthesize, simple to characterize, and can improve the future materials.

## Figures and Tables

**Figure 1 molecules-26-03063-f001:**
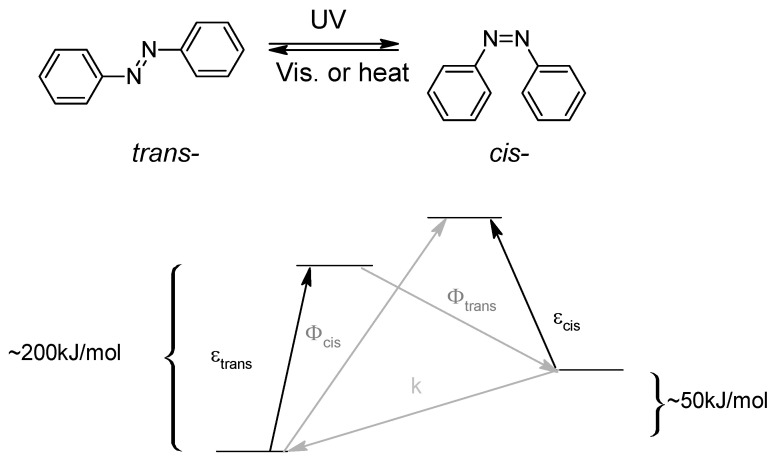
Isomerization capabilities of a nude azobenzene and simplified state model for a nude azobenzene. ε = extinction coefficients, Φ = quantum yields for the photoisomerization, k = rate of thermal relaxation.

**Figure 2 molecules-26-03063-f002:**
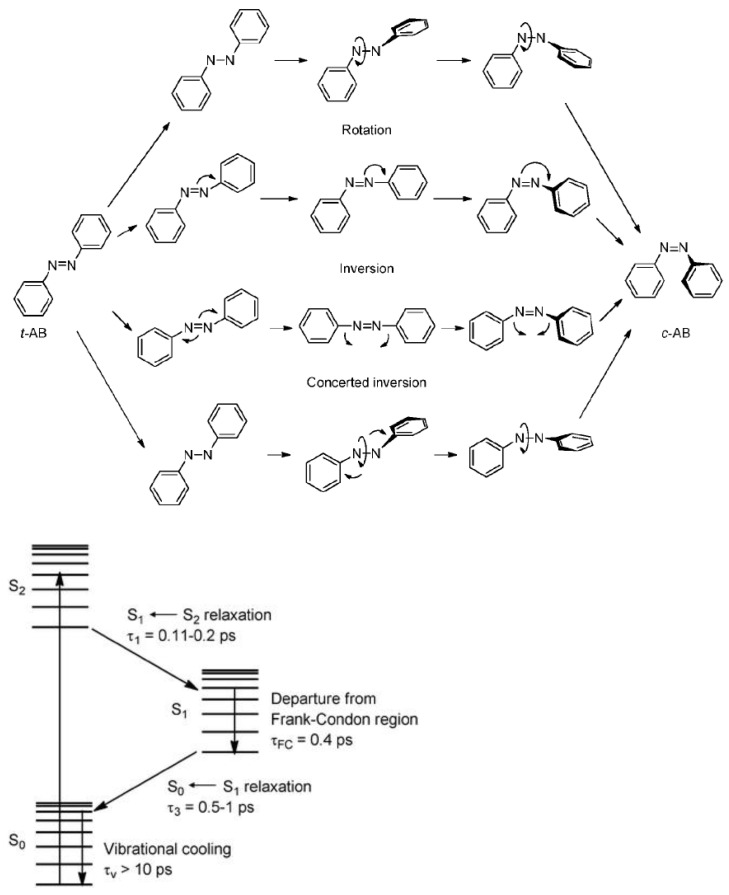
Proposed mechanisms for the trans-cis isomerization; A simplified Jablonski diagram showing the S0, S1 and S2. states of trans-azobenzene. Reuse (reprint) with permission of Bandara, H.M.D.; Burdette, S.C. Photoisomerization in Different Classes of Azobenzene. *Chem. Soc. Rev.*
**2012**, *41*, 1809–1825, doi:10.1039/C1CS15179G. Copyright (2012) Royal Society of Chemistry.

**Figure 3 molecules-26-03063-f003:**
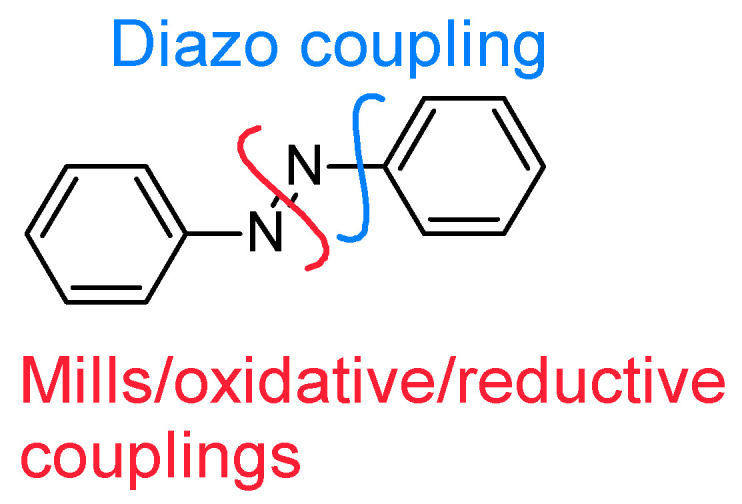
Ways of azobenzene formation.

**Figure 4 molecules-26-03063-f004:**
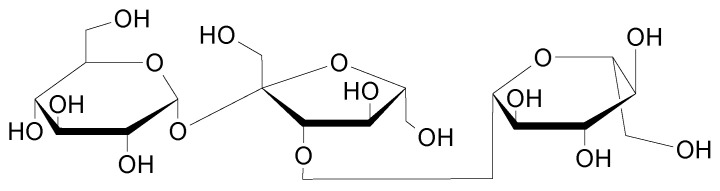
Melecitose structure.

**Figure 5 molecules-26-03063-f005:**
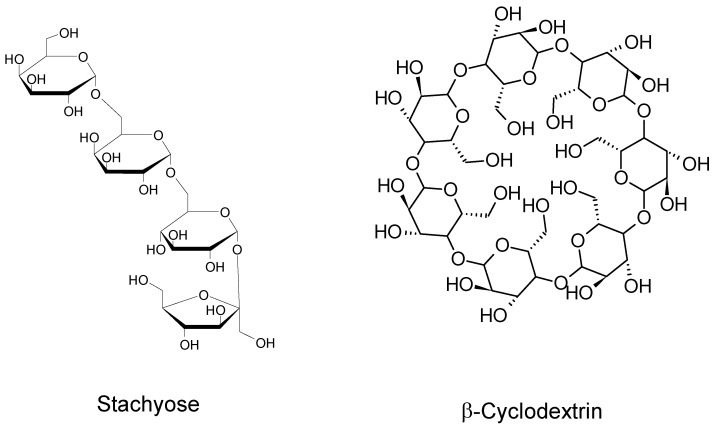
Linear and cyclic models of oligosaccharides.

**Figure 6 molecules-26-03063-f006:**
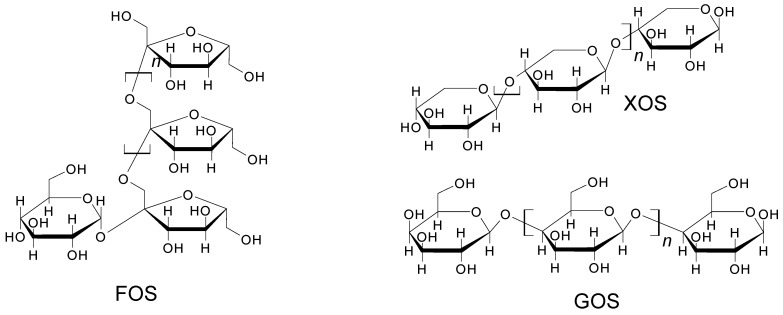
Molecular structures of FOS, XOS and GOS.

**Figure 7 molecules-26-03063-f007:**
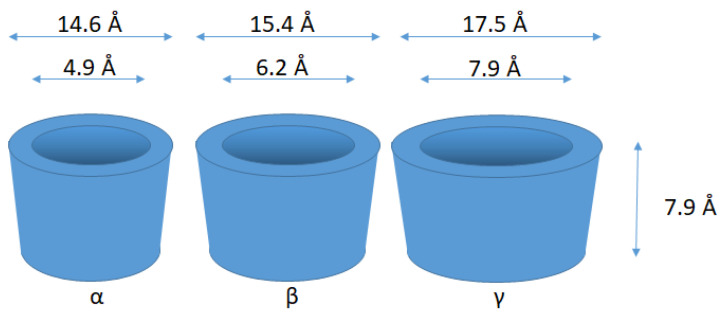
Various dimensions of cyclodextrins.

**Figure 8 molecules-26-03063-f008:**
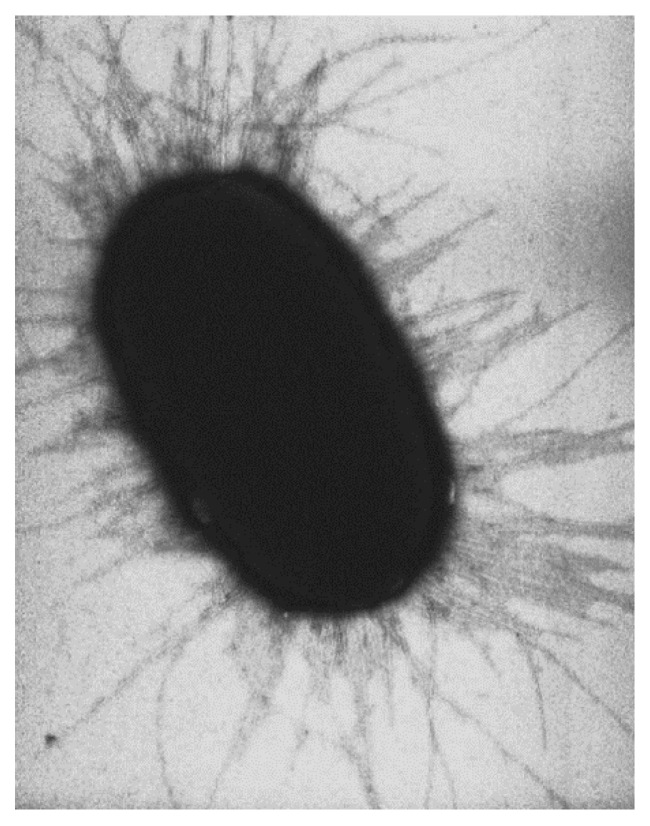
Fimbriae covering *E. coli*. Reuse permitted by the Creative Common CC BY license from Gross, L. Bacterial Fimbriae Designed to Stay with the Flow. *PLoS Biology*
**2006**, *4*, e314, doi:10.1371/journal.pbio.0040314. Image from Manu Forero, doi:10.1371/journal.pbio.0040314.g001.

**Figure 9 molecules-26-03063-f009:**
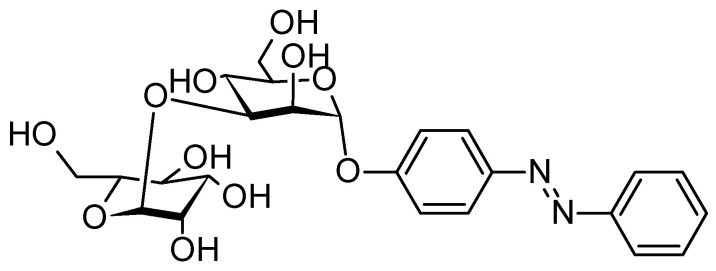
Azobenzene mannobioside.

**Figure 10 molecules-26-03063-f010:**
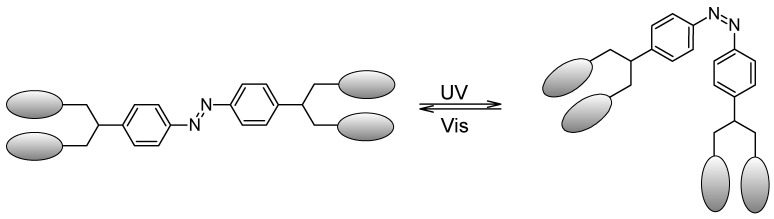
Schematic isomerization of multivalent sugar ligand azobenzene. The elliptic drawings represents the sugar moieties.

**Figure 11 molecules-26-03063-f011:**
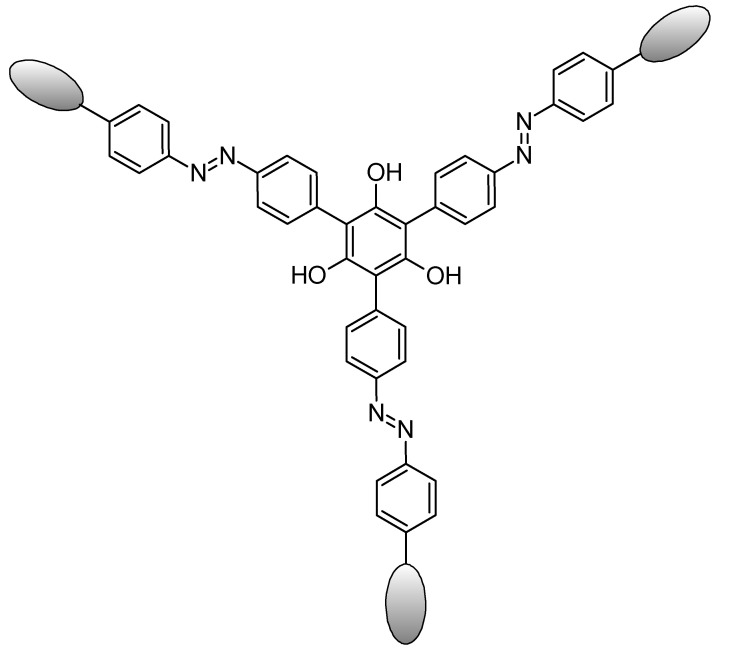
Schematic representation of phlorocello. The elliptic drawings represents the cellobiosic moieties.

**Figure 12 molecules-26-03063-f012:**
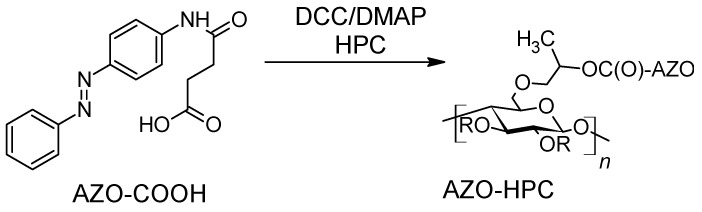
Synthetic pathway for AZO-HPC formation.

**Figure 13 molecules-26-03063-f013:**
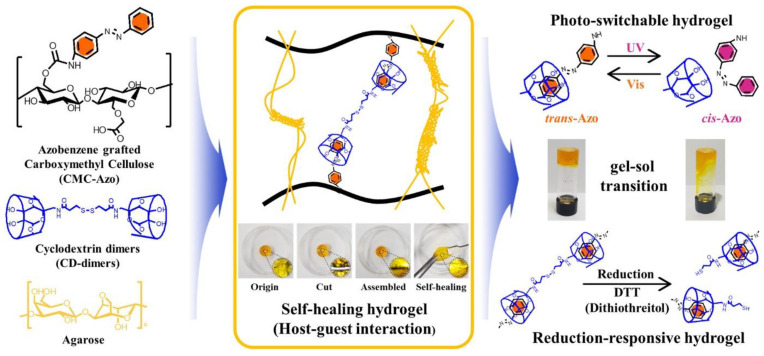
Photo-switchable, reduction-responsive, and self-healing CMC-Azo hydrogels based on host-guest interactions with CD-dimers. Reuse (reprint) with permission of Kim, Y.; Jeong, D.; Shinde, V.V.; Hu, Y.; Kim, C.; Jung, S. Azobenzene-Grafted Carboxymethyl Cellulose Hydrogels with Photo-Switchable, Reduction-Responsive and Self-Healing Properties for a Controlled Drug Release System. *International Journal of Biological Macromolecules*
**2020**, *163*, 824–832, doi:10.1016/j.ijbiomac.2020.07.071. Copyright (2016) Elsevier.

**Figure 14 molecules-26-03063-f014:**
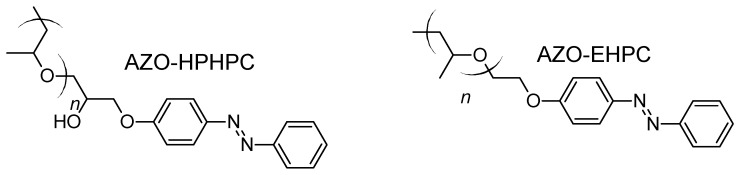
AZO-EHPC or AZO-HPHPC formula.

**Figure 15 molecules-26-03063-f015:**
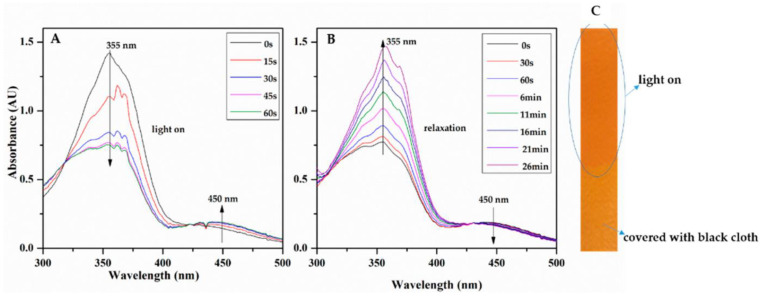
The light-responsive behavior of the MMAZO in ethanol (**A**,**B**) and the PMMAZO-graft CNCs and polyurethane composite film (**C**). Reuse permitted by the Creative Common CC BY license from Liu, X.; Li, M.; Zheng, X.; Retulainen, E.; Fu, S. Dual Light- and PH-Responsive Composite of Polyazo-Derivative Grafted Cellulose Nanocrystals. *Materials*
**2018**, *11*, 1725, doi:10.3390/ma11091725.

**Figure 16 molecules-26-03063-f016:**
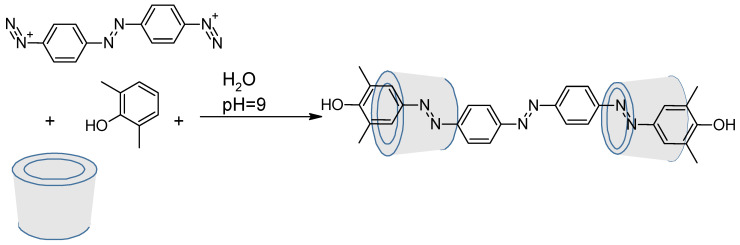
Diazo coupling inside a cyclodextrin for rotaxane synthesis.

**Figure 17 molecules-26-03063-f017:**
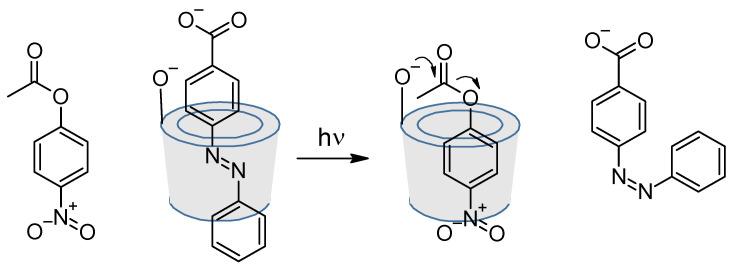
Excluding photochromic inhibitor.

**Figure 18 molecules-26-03063-f018:**
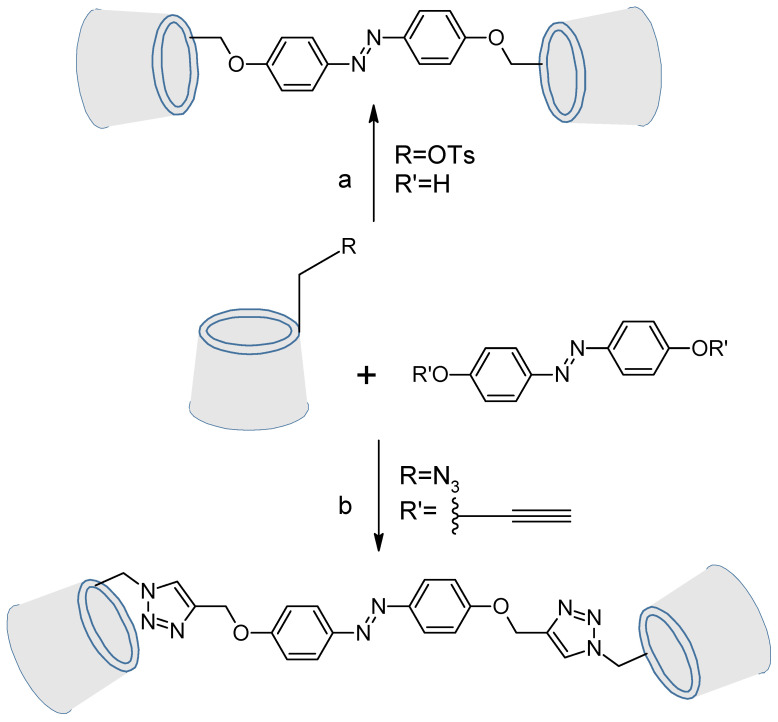
Ways of syntheses. a. Cs_2_CO_3_, DMF, 90 °C, 24 h; b. (EtO)_3_P·CuI, DMF, 100 °C, 2 h.

**Figure 19 molecules-26-03063-f019:**
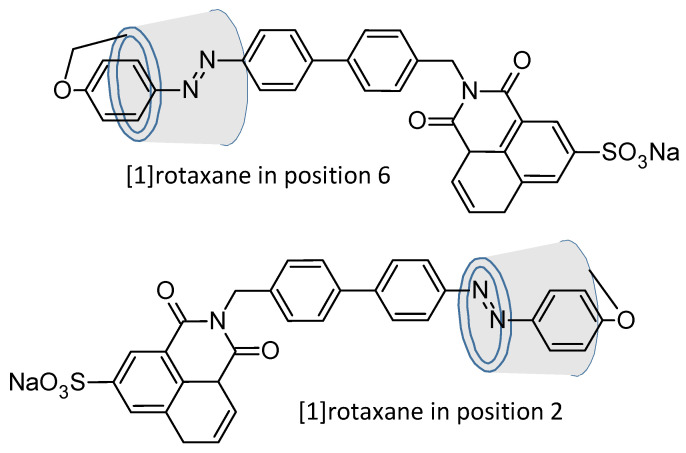
Roraxanes from Ma et al.

**Figure 20 molecules-26-03063-f020:**
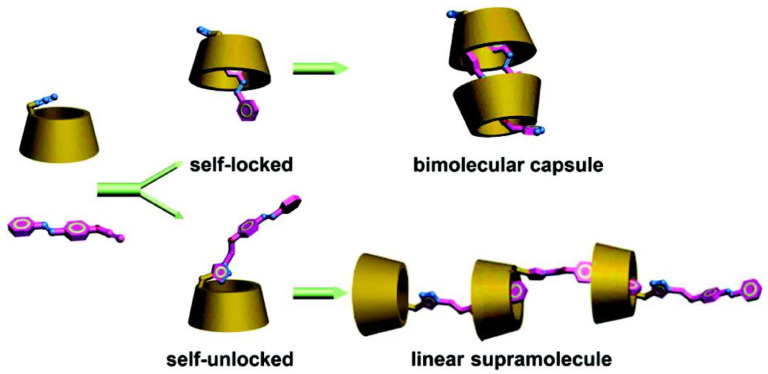
Self locking/unlocking molecules upon their way of synthesis. Reuse (reprint) with permission of Liu, Y.; Yang, Z.-X.; Chen, Y. Syntheses and Self-Assembly Behaviors of the Azobenzenyl Modified β-Cyclodextrins Isomers. *J. Org. Chem.*
**2008**, *73*, 5298–5304, doi:10.1021/jo800488f. Copyright (2008) American Chemical Society.

**Figure 21 molecules-26-03063-f021:**
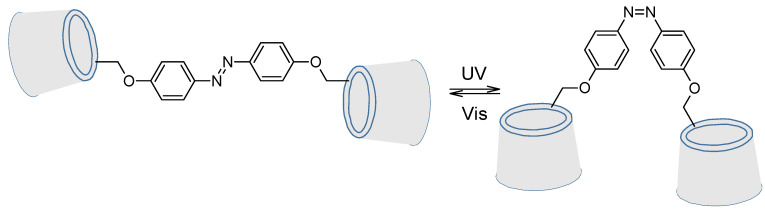
Absorption spectra of 1 before (-) and after (- - -) photoirradiation. Reuse (reprint) with permission of Aoyagi, T.; Ueno, A.; Fukushima, M.; Osa, T. Synthesis and Photoisomerization of an Azobenzene Derivative Bearing Two β-Cyclodextrin Units at Both Ends. *Macromolecular Rapid Communications*
**1998**, *19*, 103–105, doi:10.1002/(SICI)1521-3927(19980201)19:2<103::AID-MARC103>3.0.CO;2-R. Copyright (1998) Wiley.

**Figure 22 molecules-26-03063-f022:**
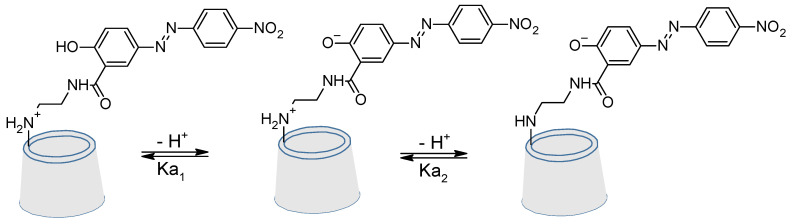
Absorption spectra at various pH values. (down) reuse (reprint) with permission of Aoyagi, T.; Nakamura, A.; Ikeda, H.; Ikeda, T.; Mihara, H.; Ueno, A. Alizarin Yellow-Modified β-Cyclodextrin as a Guest-Responsive Absorption Change Sensor. *Anal. Chem.*
**1997**, *69*, 659–663, doi:10.1021/ac960727z. Copyright (1997) American Chemical Society.

**Figure 23 molecules-26-03063-f023:**
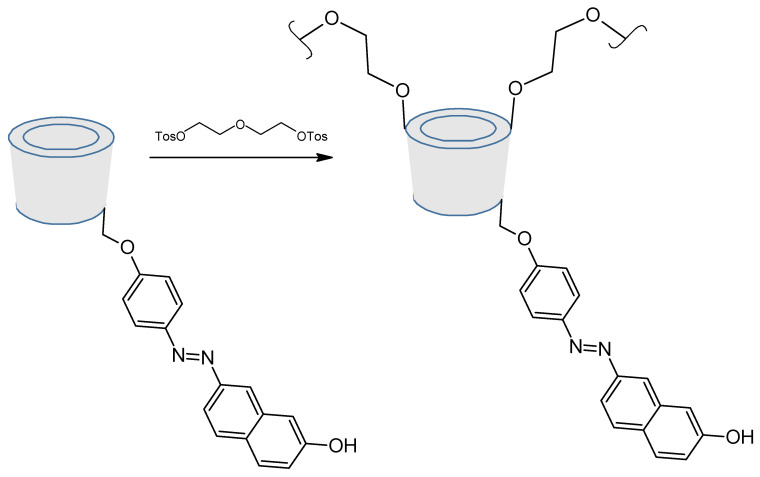
Sensor synthesis (up) and sensitivity factor against phenol, 4-chlorophenol and 2,4-dichlorophenol (down). Reuse (down) permitted by the Creative Common CC BY license from Ncube, P.; Krause, R.W.; Mamba, B.B. Fluorescent Sensing of Chlorophenols in Water Using an Azo Dye Modified β-Cyclodextrin Polymer. *Sensors*
**2011**, *11*, 4598–4608, doi:10.3390/s110504598.

**Figure 24 molecules-26-03063-f024:**
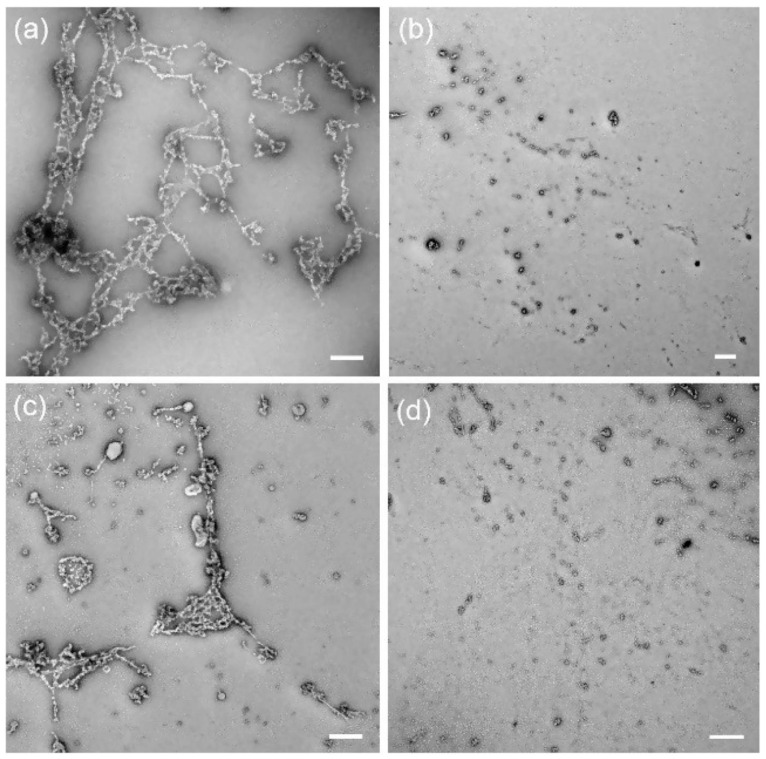
TEM images of βCD-C6-Azo aqueous solution (1 mM): (**a**) original solution; (**b**) original solution irradiated by 360 nm UV light; (**c**) original solution irradiated by 360 nm UV light and then visible light; (**d**) original solution irradiated by 360 nm UV light and then visible light and then UV light. The scale bar = 0.5 μm. Reuse (reprint) with permission of Guo, H.; Jiang, B.; Zhou, J.; Zhao, L.; Xu, B.; Liu, C. Self-Assembly of β-Cyclodextrin-Derived Amphiphile with a Photo Responsive Guest. *Colloids Surf. A Physicochem. Eng. Asp.*
**2019**, *579*, 123683, doi:10.1016/j.colsurfa.2019.123683. Copyright (2019) Elsevier.

**Figure 25 molecules-26-03063-f025:**
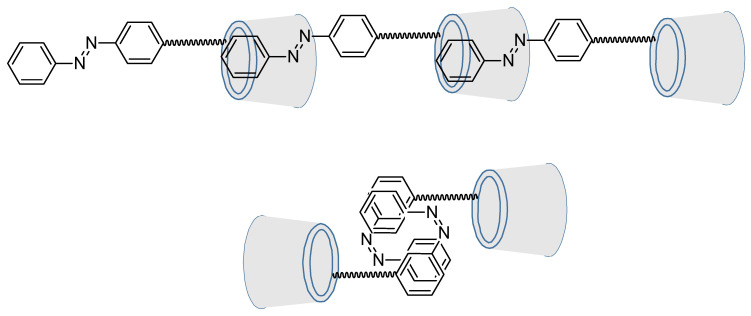
Schematic assembling of trans and cis molecules.

**Table 1 molecules-26-03063-t001:**
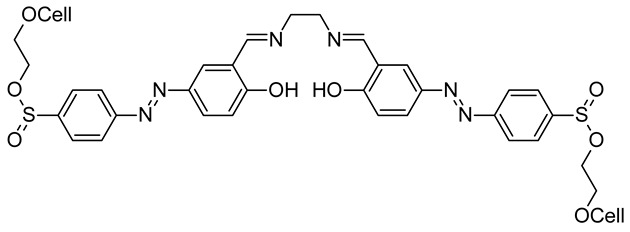
Azobenzenic Schiff bases coupled on cellulose and UV protective properties.

Samples	Concentrations (g/L)	UPF	T(UVA)	T(UVB)
**Control**	0	8.31	10.09	11.81
**FC-1**	2	18.72	4.29	5.17
**FC-2**	5	31.7	3.09	3.73

**Table 2 molecules-26-03063-t002:**
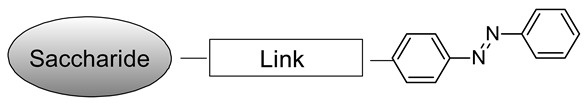
Links between azo moieties and oligosaccharides and reagents used to obtain these links.

Link	Main Reagents	Reference
Alkyle	cellulosenanocrystals-bromo-isobutyric ester, azohexyl methacrylate, CuBr, AsAc, PMDETA	[95]
Amide	glycine-azobenzene, lacto-malto-cellobiono-lactone	[83]
azobenzenecarbonyl chloride, aminoethylglycopyranosides	[84]
carboxymethylcellulose, aminoazobenzene, EDC	[90]
azobenzenedicarboxylic acid, aminoethylaminodeoxycyclodextrin, DCC, HOBt	[114]
azobenzeneoxypentylcarbonyl chloride, aminoethylaminodeoxycyclodextrin	[115]
Ester	bis-chlorocarbonylazobenzene, cyclodextrin, pyridine	[111]
Ether	mannosyl trichlororacetimidate, hydroxyazobenzene, BF_3_.Et_2_O	[82]
bromoethoxyazobenzene, hydroxypropylcellulose, Na_2_CO_3_, KI	[92]
cellulose, bis-sulphatoethylsulfonylazobenzene, Na_2_CO_3_	[96]
hydroxyazobenzene, O-tosyl-cyclodextrin, K_2_CO_3_	[105]
dihydroxyazobenzene, O-tosyl-cyclodextrin, NaOH	[107]
dihydroxyazobenzene, O-tosyl-cyclodextrin, K_2_CO_3_	[112]
hydroxynaphtaleneazophenol, O-tosyl-cyclodextrin, K_2_CO_3_	[113]
Oxapentylamine	naphtylsulfonylcyclodextrin, azobenzenoxy-oxapentylamine, DMF	[110]
Triazole	propargyloxyazobenzene, azidocyclodextrin, [ethanol, water] or [CuSO_4_·5H_2_O, sodium ascorbate]	[106]
Ether with in situ diazotation	diazotation from aminophenyl cellobioside, phloroglucinol	[85]

**Table 3 molecules-26-03063-t003:** Summary of the various applications for azo-guest or azo-grafted oligosaccharides.

Application	Reference (Year of Publication)
Inhibitory of cell surface adhesion, immunochemistry, UV protection	82 (**2013**), 83 (**2012**), 84 (**2005**), 85 (**1965**), 96 (**2012**)
Hydrogels, liquid/nano crystals	90 (**2020**), 92 (**2016**), 95 (**2018**)
Smart rotaxanes and self-assemblies	102 (**1999**), 105 (**2007**), 106 (**2008**), 115 (**2019**)
Sensors	110 (**1991**), 111 (**1979**), 112 (**2004**), 113 (**2011**), 114 (**2002**)

## Data Availability

Review paper, no data available.

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
