# Peer review of "Azo-Dyes-Grafted Oligosaccharides—From Synthesis to Applications"

_molecules, 2021, doi:10.3390/molecules26113063_

Round 1

Reviewer 1 Report

The work lacks a more detailed and in-depth description and explanation
of the process that the text deals with (photochromic behavior). It needs to be much more detailed,
at the level of electron level configuration, how molecules behave and how
they interact with photons. Beside that, principal cis-trans isomerization
should be discussed and explained in general level.
Also, more detailed applications should be discussed,
e.g. I. Kratochvílová, et al:
Organic FET-Like Photoactive Device, Experiments and DFT Modeling, 
European Physical Journal E 25 (2008) 299-307.

Reviewer 2 Report

The manuscript submitted Léonard and Fayeulle is about a detailed review of azo-dyes-grafted oligosaccharides and their applications. This work is interesting in the field and the manuscript is well structured, however, there are some items that need to be addressed before this work is suitable for publication:

-           Authors should present a table that summarize the different techniques synthesis to obtain azo-dyes-grafted oligosaccharides, highlighting the most representative works in the last years. In the same sense, a table about the applications should be also included, highlighting the main contribution of each work.

Round 2

Reviewer 1 Report

To be published.

Reviewer 2 Report

I consider that this revised manuscript is suitable for publication.